# Barkhausen Noise Emission as a Function of Tensile Stress in Low-Alloyed Steels: Influence of Corrosion and Steel Strength

František Bahleda [1], Miroslav Neslušan [2,*], Filip Pastorek [3], Radoslav Koňár [2] and Tibor Kubjatko [4]

1   Faculty of Civil Engineering, University of Žilina, Univerzitná 1, 01026 Žilina, Slovakia; frantisek.bahleda@uniza.sk
2   Faculty of Mechanical Engineering, University of Žilina, Univerzitná 1, 01026 Žilina, Slovakia; radoslav.konar@fstroj.uniza.sk
3   Research Centre, University of Žilina, Univerzitná 1, 01026 Žilina, Slovakia; filip.pastorek@rc.uniza.sk
4   Institute of Forensic Research and Education, University of Žilina, Univerzitná 1, 01026 Žilina, Slovakia; kubjatko@uniza.sk
*   Correspondence: miroslav.neslusan@fstroj.uniza.sk; Tel.: +421-908-811-973

**Abstract:** Components of civil buildings are frequently made of low-alloyed steel, which can suffer from corrosion damage. This damage reduces their bearing capacity and/or redistributes the load to neighboring regions with the potential risk of their collapse. For this reason, this study deals with the non-destructive monitoring of bodies made of these steels based on Barkhausen noise emission. The superimposing contribution of corrosion extent and exerted tensile load is investigated on samples of variable yield strength in the range from 235 to 1100 MPa. It is found that the presence of a corroded layer attenuates Barkhausen noise and makes the body harder from a magnetic point of view. The reduced effective thickness of the samples as a result of corrosion damage increases the true stress. Barkhausen noise grows along with the tensile stress in the direction of exerted stress at the expense of decreasing Barkhausen noise in the perpendicular direction. The evolution of Barkhausen noise versus tensile stress is mostly shifted to the lower values of Barkhausen noise, along with the increasing degree of corrosion damage. The evolution of Barkhausen noise versus tensile stress is also affected by the initial microstructure and the corresponding yield strength of the low-alloyed steel. Corrosion attack results in the growth of *FWHM*, which is compensated by the decreasing evolution along with the tensile stresses. The effective values drop down with the higher extent of corrosion damage. However, the response with respect to the tensile stress is asymmetric in RD and TD due to the realignment of DWs into RD. Finally, *PP* tends to increase with the corrosion attack as well as the tensile stress and this parameter only exhibits the systematic behavior in RD as well as TD. On the other hand, MBN-extracted parameters as well as their combination provide no exclusive values on which the pure contribution of corrosion and tensile stress can be distinguished.

**Keywords:** Barkhausen noise; low-alloyed steels; corrosion; tensile stress

## 1. Introduction

Low-alloyed steels are frequently employed in the construction of bridges and civil structures, in the automotive industry, and in other applications. Their mechanical properties strongly depend on the microstructure and the corresponding thermomechanical treatment [1–4]. Low-alloyed steels with higher yield and ultimate strength are mostly due to grain refinement and/or phase transformation, mostly developed after hot rolling. The yield strength of low-alloyed steels up to 700 MPa can be obtained via grain refinement, and these steels are mostly composed of ferritic grains of reduced size [1,2]. The further growth of yield strength can be produced via accelerated cooling rates and the corresponding bainite or martensite microstructure (or their mixture). Their alloying is quite limited, and a certain fraction of Ti or/and Nb is added to strengthen the matrix (this contributes to grain refinement as well as matrix precipitation hardening) [1,2]. The high-strength

low-alloyed steels are considered cheap materials. Obtaining good mechanical properties despite increasing strength is at the expense of decreased toughness [2,3].

The functionality of components made of these steels can be negatively affected by corrosion, which can remarkably damage surface properties and/or reduce the bearing capacity with respect to decreasing the effective cross-section. Furthermore, increasing the true stress in components as a result of corrosion can redistribute stress to neighboring regions [5,6]. Yielding components develop unacceptable deformations of construction, whereas plastic instability can initiate premature failures and collapse [5–8]. To reveal the critical state through visual observation can be a risky approach since many defects are hidden and cannot be found by the naked eye. For these reasons, many non-destructive techniques have been developed in order to monitor the extent of corrosion and its influence on construction stability. These techniques are based on a diversity of physical phenomena such as acoustic emission [9] (cluster analysis applied for post-processing), ultrasonic tomography [10], gamma rays [11], or electromagnetic signals [12]. These techniques have found industrial relevance in some particular cases and can be employed for reliable inspections of bodies subjected to environmental damage (especially corrosion).

In addition, magnetic methods, especially those based on magnetic inductive testing [13], can be applied for the aforementioned purpose. Moreover, Barkhausen noise emission (MBN—magnetic Barkhausen noise) was reported as a potential technique for the assessment of corrosion extent in wires [14] or conventional flat bodies [15,16]. The origin of MBN can be found in electromagnetic pulses produced by magnetic domain walls (DWs) during their irreversible and discontinuous motion under an altering magnetic field [17–19]. MBN is strongly affected by three aspects: microstructure, surface topography, and stress state. All lattice imperfections (dislocations, grain boundaries, precipitates, hard and non-ferromagnetic phases, etc.) pin DWs in their position and/or interfere with DWs in motion [20–23]. This aspect contributes to decreasing MBN. Increasing surface irregularities make the propagation of MBN pulses toward a sensing coil more difficult, which also attenuates MBN. With respect to the stress state, it should be distinguished between external stresses and the residual state. It is well known that tensile stresses tend to align DWs in the direction of this stress, whereas compressive stresses align DWs perpendicularly against the compressive stress (valid for Fe alloys). Therefore, tensile stress usually increases MBN, whereas compressive stress lowers MBN [24–26]. The influence of residual stress is less pronounced, especially in Fe alloys with a complicated microstructure [19,27]. On the other hand, the relationship between external stress and MBN is much closer [24,28]. The sensitivity of MBN against external stress depends on many variables, and early saturation can be found in some cases [29].

This paper represents a further study in which MBN is investigated as a function of stress state and superimposing corrosion damage. The pilot studies in this field were related to monitoring the corrosion extent only [14,16], whereas the contribution of corrosion to the relationship between MBN and tensile stress was reported for the low-alloyed fully ferritic steel MC460 [15]. This study investigates the influence of variable strength and the corresponding microstructure on the relationship between MBN and tensile stresses. As contrasted against the previous study [16], three different grades of the low-alloyed steels are compared, such as fully ferritic steel S235, MC700 (a mixture of ferrite and bainite), and finally, fully martensitic steel MC1100. The evolution of MBN and the extracted features are analyzed with respect to different corrosion damages. This study represents a second report following up on the previous one [16] in which the corroded samples were loaded by uniaxial stress. MBN as a monitoring technique is very fast and reliable. Previous experience demonstrated its good sensitivity against the developed corrosion extent [16]. Furthermore, MBN is sensitive to the stress state [24,28]. For these reasons, this technique is employed in this study for the aforementioned purpose.

## 2. Experimental Conditions

The investigations were conducted on the three different low-alloyed steels of nominal yield strength: 235, 700, and 1100 MPa. Twenty-four samples of each steel (3 repetitive samples for 8 different corrosion attacks), 200 mm long, 20 mm wide, and 5 mm thick, were cut along the rolling direction (RD) of the delivered sheet thickness of 5 mm. The thickness of samples made of MC1100 was 6 mm. The sheet of yield strength 235 MPa represents a conventional low-strength low-alloyed steel that is widely used and commercially available. The sheets with yield strengths of 700 and 1100 MPa were produced by hot rolling and consecutive accelerated cooling. The microstructure of MC700 is a mixture of ferrite and bainite. Apart from ferrite grain refinement and the presence of the bainite phase as the major aspects linked with high strength, the matrix is also strengthened by Nb and Ti precipitates [1,2]. The microstructure of MC1100 is fully composed of martensite without additional strengthening by Nb or Ti. The martensite microstructure is due to an accelerated cooling rate after hot rolling, which is higher than that of MC700. The chemical composition, as well as the microstructure images of these steels, can be found in the previous study [16].

The true mechanical properties were verified by the standardized uniaxial tensile test, as illustrated in Figure 1, on dog-bone-shaped samples (gauge length of 40 mm and width of 12 mm). The true elastic strains were checked by the use of an Instron dynamic strain gauge extensometer 2620-602 on the length of 25 mm. The tensile tests, as well as the loading of the samples after corrosion attack, were performed using an Instron 5985 device. The true yield strength of S235 was $312 \pm 6$ MPa, the ultimate strength was $406 \pm 7$ MPa, and the elongation at break was $39 \pm 1.5\%$ (obtained from three repetitive measurements). The true yield strength of MC700 was $764 \pm 23$ MPa, the ultimate strength was $829 \pm 12$ MPa, and the elongation at break was $24 \pm 1.2\%$. The true yield strength of MC1100 was $1232 \pm 15$ MPa, the ultimate strength was $1333 \pm 12$ MPa, and the elongation at break was $12 \pm 0.6\%$.

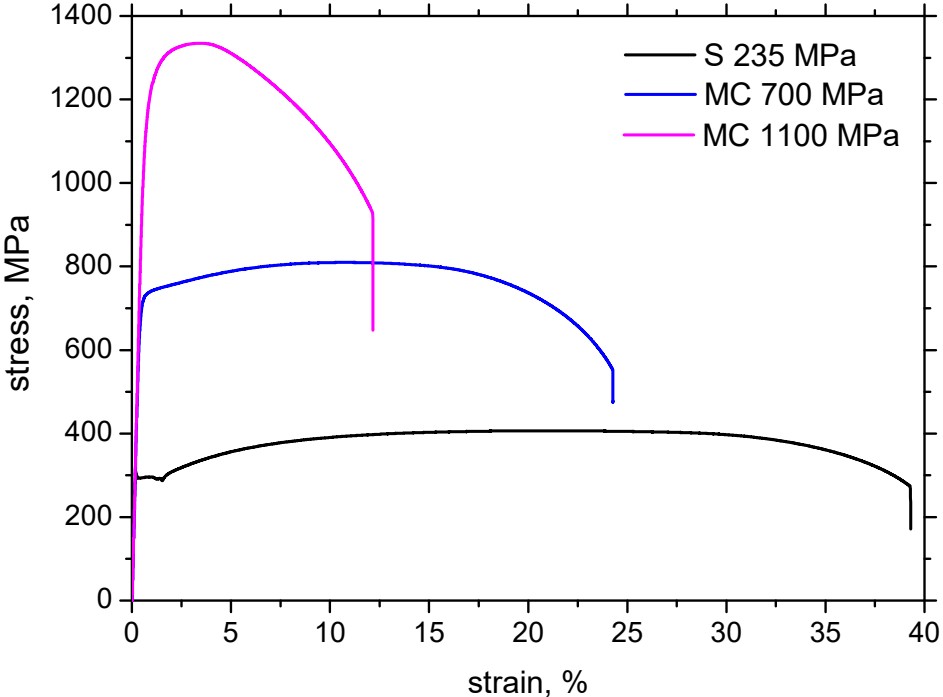

**Figure 1.** Engineering stress–strain curves of investigated alloys.

The samples were subjected to corrosion attack, which varied in terms of the number of days in the corrosion chamber (see Table 1). The specimens were artificially corroded in a neutral salt spray atmosphere (in accordance with the STN EN ISO 9227 standard) in a VSC KWT 1000 corrosion chamber (temperature of 35 °C, pH in the range from 6.5 to 7.2,

pressure in the chamber of 120 kPa). The specimens were cleaned before the experiments and placed into the chamber with the use of plastic strips. For each steel, one sample without corrosion attack was measured as well (as received state). Loose corrosion residues were manually removed from the surface of each sample by a wire brush, and subsequently, the surface of the specimens was water-cleaned and dried.

**Table 1.** Number of days in the corrosion chamber.

| Sample Number | 1 | 2 | 3 | 4 | 5 | 6 | 7 | 8 |
|---|---|---|---|---|---|---|---|---|
| Number of days | 1 | 6 | 11 | 18 | 27 | 41 | 55 | 69 |

In order to analyze the wall thickness as a result of corrosion attack, 20 mm long pieces were cut along RD after experiments by the Struers Secotom-50. The specimens were taped with tape in order to avoid delamination of the corroded layer from the surface during cutting. The cut specimens were hot-molded, ground, polished, and finally etched with 3% Nital. The wall thickness was observed and measured in five different positions using the light microscope Zeiss AxioCam MRc5 in Olympus SZx16 and Quick Photo Industrial 3.0 software.

The MBN signal was acquired using a RollScan 350 in situ of the tensile test. The tensile stress was increased in steps of 50 MPa up to 300 MPa, followed by increments of 100 MPa afterward. MBN was measured within the elastic region (on the stress–strain curves) along the length of the sample (RD) as well as in the transversal direction (TD) using a serial sensor S1-18-12-01. Measurements of MBN in two perpendicular directions were carried out by magnetizing pole rotation. MBN signals were recorded and filtered in MicroScan 600 software under a magnetizing voltage of 16 V and a sine profile frequency of 125 Hz. The sensor during the tensile test was fixed on the sample surface with a spring in order to maintain constant conditions for the transmission of electromagnetic pulses toward the sensing coil (for further details and a brief illustration, please check the study in [30]). The RD direction represents the easy axis of magnetization of the delivered sheets, whereas TD is the perpendicular hard axis. The MBN signal is composed of electromagnetic pulses in the frequency range from 10 to 1000 kHz. Three magnetic parameters were extracted from acquired MBN signals such as the following:

-    MBN, which refers to the effective value of the signal;
-    *PP* (Peak Position), which is linked to the position of the maximum of the MBN envelope;
-    *FWHM* (full-width at half-maximum) of the MBN envelope.

A brief diagrammatic sketch of equipment and extracted information for easier navigation through the paper is presented in Figure 2.

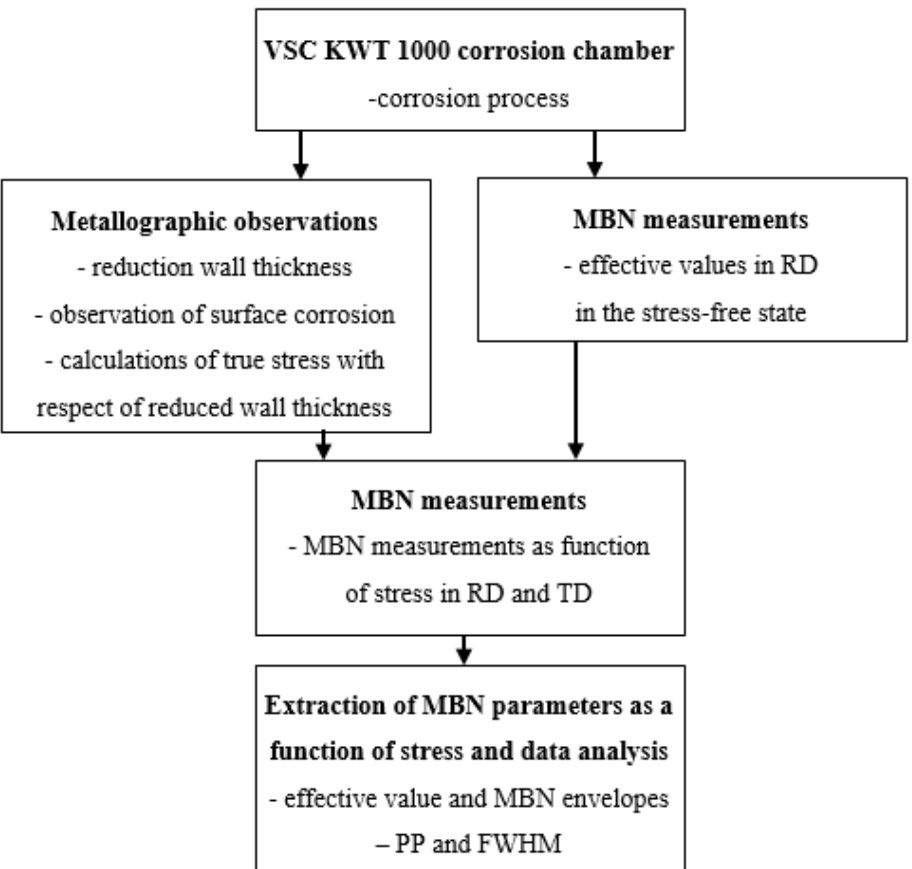

**Figure 2.** The brief diagrammatic sketch of equipment and extracted information.

### 3. Results of Experiments and Discussion

*3.1. Wall Thickness*

Figure 3 illustrates two different aspects of corrosion attack. The first one is associated with the increasing height of surface irregularities. The surface becomes rougher as a result of the heterogeneity of corrosion attack. The electrochemical process in the neighboring regions makes certain regions relatively anodic or cathodic to the neighboring ones. Moreover, this anodic-to-cathodic relationship can be exchanged as a result of the different elemental compositions. For this reason, the uncorroded regions previously unaffected by corrosion can be attacked. For these reasons, corrosion attack is strongly heterogeneous, which in turn contributes to the variable surface height, as depicted in Figure 3. It was previously reported that progressively developed corrosion attack remarkably increases *Sa*, *Sq*, and *Sz* [15].

The second aspect of corrosion attack is associated with a reduction in the effective wall thickness (see Figures 3 and 4). Figure 4 clearly demonstrates that the effective wall thickness is progressively reduced, and the gradient of this reduction is kept nearly constant within the investigated duration of the corrosion attack. Moreover, Figure 4 also illustrates that the reduction in the effective wall thickness is less for MC1100 when contrasted against MC700 and S235 steels. This finding proves that the resistance against corrosion in the case of high-strength low-alloyed steels is better than that of lower-strength steels [1,2]. This information also proves the previous study [16], which clearly demonstrates that the thickness of a corroded layer on an un-corroded matrix is highest for S235, lower for MC700, and lowest for MC1100.

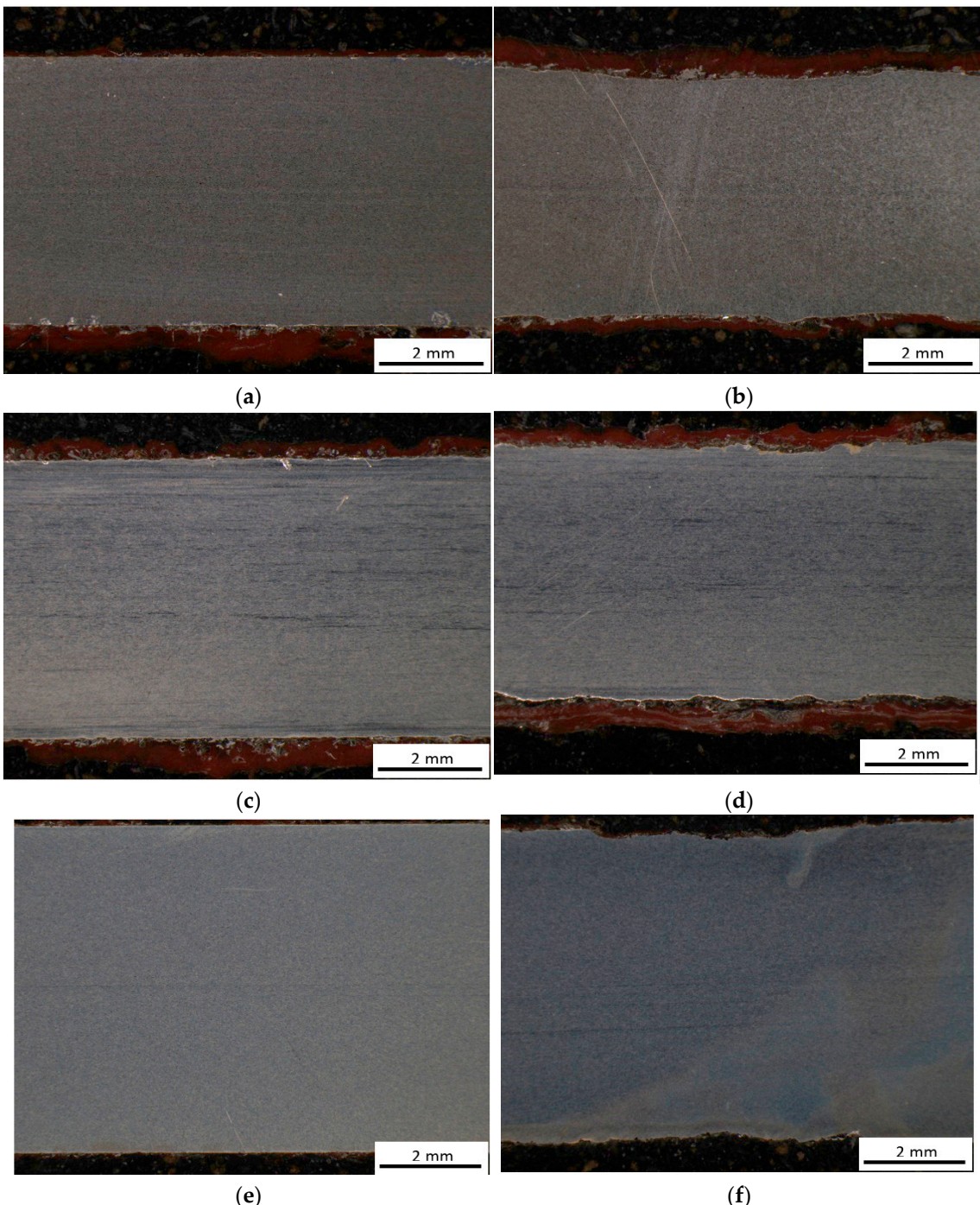

**Figure 3.** Cross-sectional views of the steels before and after corrosion attack. (**a**) S235—bulk, (**b**) S235—after 69 days, (**c**) MC700—bulk, (**d**) MC700—after 69 days, (**e**) MC1100—bulk, (**f**) MC1100—after 69 days.

Comparing the evolution of the wall thickness reduction (see Figure 4) and the evolution of the thickness of the corroded layer, some differences can be found. The differences among the steels with respect to the thickness of the corroded layer are more pronounced. The evolution of the effective wall thickness, along with the number of days in the chamber, is different. This evolution exhibits continuous growth with a nearly constant gradient, whereas the thickness of the corroded layer increases remarkably within the initial phases of the corrosion attack, followed by its saturation [16]. An explanation of this controversy can be found in the removal of poorly adhered corrosion residues from the surface af-

ter a corrosion attack. Corrosion damage penetrates deeper and deeper along with the prolonged time period within the matrix exposed to the aggressive salt bath. On one hand, the corroded surface layer, which remains on the surface after cleaning, saturates in thickness, whereas the thickness of the corroded layer being removed during the cleaning continuously grows.

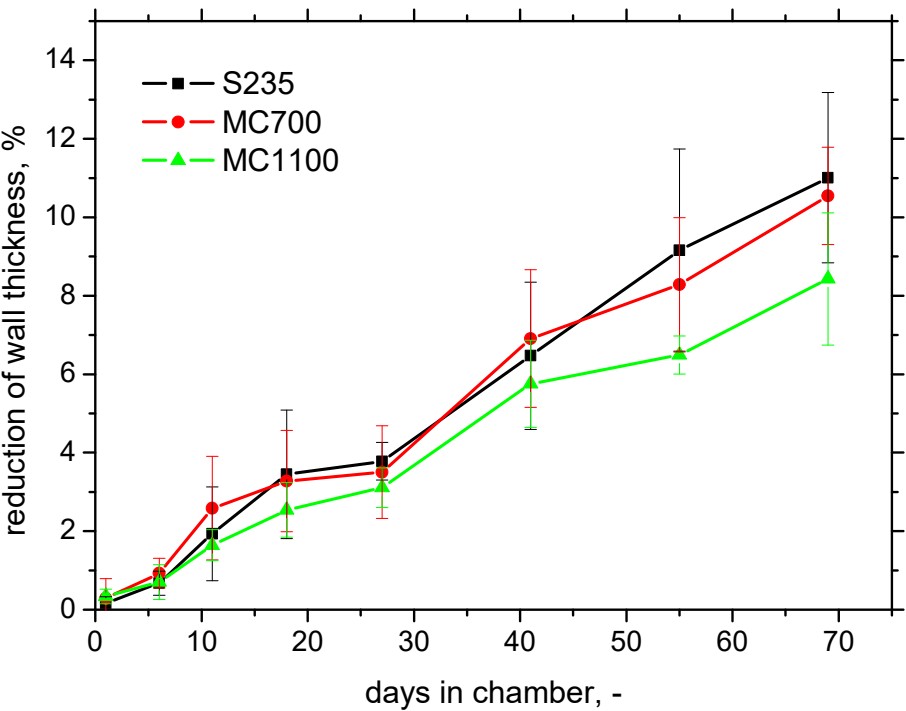

**Figure 4.** Reduction in wall thickness as a function of days in chamber.

Standard deviations of the measured effective wall thickness increase along with the prolonged corrosion attack as a result of the increasing height of surface irregularities. Reduced wall thickness should be taken into account with respect to the true stresses exerted during uniaxial tensile stressing of the corroded samples. For this reason, the information in Figure 4 is employed for the calculation of true stresses, as analyzed in the next chapter. Metallographic details in which the corroded layer can be distinguished from unaffected matrix can be found in the previous study [16]. SEM observations as well as the chemical analyses of the corroded layer have also been reported earlier [14].

### 3.2. MBN Measurements

It has already been reported that corroded bodies emit lower MBN [14–16]. The physical background of this behavior has been explained as a result of different effects and their superimposing contributions. A brief list can be summarized as follows:

- The corroded layer on the surface makes a weaker magnetic field in the un-corroded matrix;
- The corroded layer attenuates electromagnetic pulses propagating toward the free surface;
- Increasing the height of surface irregularities also makes the transfer of electromagnetic pulses toward the sensing coil more difficult [31];
- Finally, the corroded layer usually contains un-corroded fragments whose size drops down along with prolonged corrosion attack [14,15].

Figures 5–7 prove the aforementioned behavior. MBN in RD and TD drops down along with the longer time in the chamber. MBN in RD is mostly greater as compared with TD. It is only in the region of the lower magnitudes of the tensile stresses that MBN values are more balanced and the magnetic anisotropy expressed in terms of MBN is lost.

The evolution of MBN with respect to different yield strengths is not systematic. MBN for S235 is greater when compared with MC700, and MBN for MC700 is less than that for MC1100. On the other hand, it can be seen that MBN in TD for MC700 is more than that for MC1100 at the higher tensile stresses, especially for the heavily corroded samples (see Figures 6b and 7b). The main reason for this can be viewed in the near-surface region, which can be more or less affected by the hot rolling process and cooling conditions, as discussed in the previous study [16]. The valuable heterogeneity in the microstructure with respect to the different layers beneath the free surface is due to the hot rolling process, and this heterogeneity is most apparent in the case of MC700 [16]. The grain size in the near-surface with a thickness of about 0.1 mm is larger when contrasted against the deeper regions, and the hardness is less as a result of thermal softening. MBN of the low-alloyed steels is very often driven by grain size and the corresponding density of DWs [32]. For this reason, MBN can be expressed as follows:

$$\text{MBN (rms)} = C_g \, d^{-1/2} \tag{1}$$

where $C_g$ is the grain size and ($d$) is a dependent constant. Having in mind Equation (1), it can be easily understood that a rougher grain size results in a lower MBN.

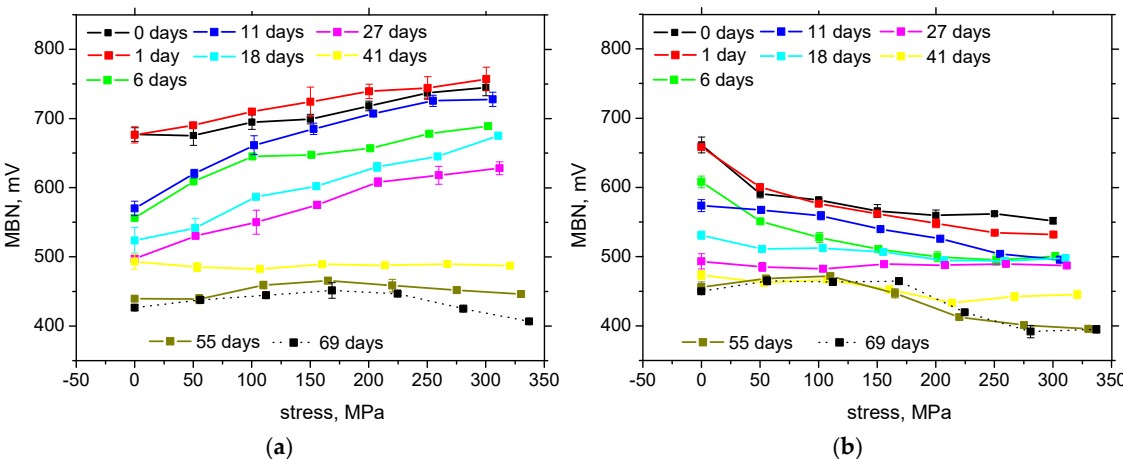

**Figure 5.** Evolution of MBN versus tensile stress for S235. (**a**) RD, (**b**) TD.

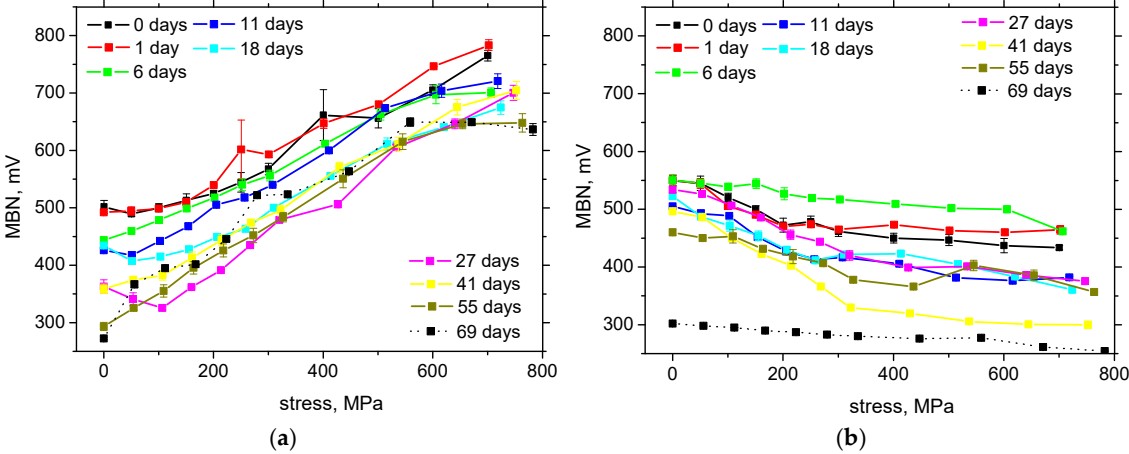

**Figure 6.** Evolution of MBN versus tensile stress for MC700. (**a**) RD, (**b**) TD.

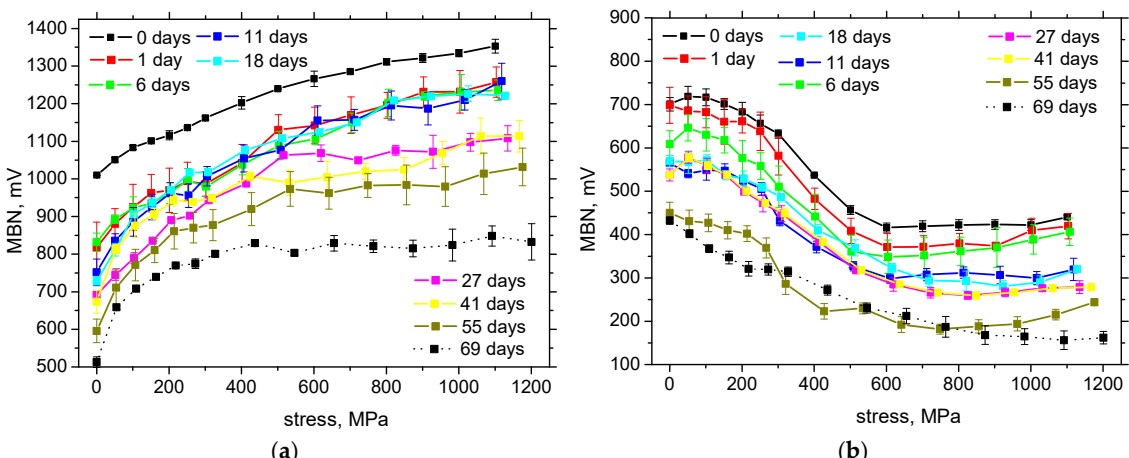

**Figure 7.** Evolution of MBN versus tensile stress for MC1100. (**a**) RD, (**b**) TD.

Figures 5–7 demonstrate that MBN grows along with increasing tensile stresses at the expense of decreasing MBN in TD. This is a result of magneto-elastic coupling and the positive magnetostriction of Fe alloys [17,19]. The increase in MBN in RD (along the direction of tensile stress) is due to the predominance of the energy of magnetocrystalline anisotropy, $E_a$, over the magnetoelastic energy, $E_\sigma$. $E_a$ depends on the magnetocrystalline anisotropy, and for the bodies of the *bcc* lattice, it can be expressed as follows [17–19].

$$E_a = K_1(\alpha_1^2\alpha_2^2 + \alpha_2^2\alpha_3^2 + \alpha_1^2\alpha_3^2) + K_2(\alpha_1^2\alpha_2^2\alpha_3^2) \tag{2}$$

where $\alpha_1$, $\alpha_2$, and $\alpha_3$ are the direction cosines of the magnetization vector with respect to three cube edges, while $K_1$ and $K_2$ represent magnetocrystalline anisotropy constants.

$E_\sigma$ strongly depends on magnetostriction and can be expressed as follows [17–19]:

$$E_\sigma = (-3\lambda_s cos^2\varphi)/2 \tag{3}$$

where $\lambda_s$ is the isotropic magnetostriction and $\varphi$ defines the angle between the direction of magnetization and the direction of the exerted stress $\sigma$.

The aforementioned and explained evolution of MBN versus the tensile stress for S235 is valid for all applied elastic stresses. This evolution is nearly the same for the samples being corroded for shorter times (up to 27 days in the chamber) (see Figure 5a). The evolutions are shifted to lower MBN. When the corrosion attack is prolonged, the sensitivity of MBN is lost, and the relationship between MBN and tensile stress becomes flat (see Figure 5a). In addition, the decreasing evolution of MBN versus stress for S235 in TD is apparent for the initial phases of the corrosion attack. However, the lack of sensitivity appears earlier with respect to the number of days in the chamber (see Figure 5b).

The flat evolution as that depicted for S235 cannot be found in MC700 as well as in MC1100 (see Figures 6 and 7). Apart from the last sample (69 days in the chamber), all samples exhibit increasing MBN in RD compensated by the decreasing MBN in TD. The increase in MBN in RD and the decrease in TD is more or less constant for MC700 (excluding some evolutions in TD for a longer duration of corrosion attack). On the other hand, the evolutions of MBN in RD for MC1100 exhibit a decreasing gradient, which is more pronounced for the samples being corroded for longer times in the chamber. Heavily damaged samples exhibit a knee in the evolution when the gradient of the increase in MBN drops down. Moreover, a lack of sensitivity can be reported for the samples corroded for 69 days in the chamber beyond the knee (see Figure 7a). The aforementioned knee can be found at a stress of about 400 MPa and appears for all evolutions in TD as well (see Figure 7b) when the decreasing MBN beyond this threshold is replaced by saturation for higher tensile stresses. The lack of sensitivity in TD for higher stresses and/or the decreasing

gradient in RD indicates an altering ratio between $E_a$ and $E_\sigma$ when these energies become more balanced.

The lack of sensitivity for the evolution of MBN versus tensile stress for the samples being corroded for a longer time in the case of S235 is due to the remarkably higher thickness of the corroded layer on the surface, as reported earlier [16]. The corroded layer on the surface does not bear the tensile stress and acts as a barrier attenuating MBN pulses originating from the un-corroded body. As soon as the corroded layer exceeds the critical thickness with respect to the MBN sensing depth [33,34], MBN pulses originating from the un-corroded fragments in the corroded layer shadow the contribution of the electromagnetic pulses originating from the uncorroded body. The valuable lower thickness of the corroded layer for MC700 and MC1100 is linked to the later saturation of the evolution of MBN versus stress as contrasted against S235.

Finally, it should be mentioned that the heavily corroded samples at the level of the highest nominal elastic stress are yielded since the true stress exceeds the yield strength (valid for S235 and MC700) due to the reduction in the effective wall thickness. For this reason, saturation or the decrease in MBN can be found at the end of the tensile stress due to increasing dislocation density.

*PP* values usually correlate with the mechanical hardness of a body [27]. For this reason, *PP* grows along with yield strength and the corresponding hardness of the low-alloyed steel. Corrosion on the surface increases *PP* since the sample behaves as magnetically harder, but this evolution saturates all alloys in RD as well TD. *PP* for TD is higher as compared with TD since TD is the hard axis of magnetization. The evolution of *PP* versus tensile stress is quite similar for all investigated alloys, and Figure 8 illustrates an example of the evolution of *PP* with respect to corrosion attack and superimposing tensile stress. RD exhibits a better sensitivity against corrosion attack as compared with TD. An initial gentle decrease in *PP* for lower stress is followed by a gentle increase for higher stresses, but the evolutions are quite flat, and their sensitivity, especially against tensile stress, is limited. On the other hand, Figure 8a depicts that the differences among *PP* become more valuable under the tensile stresses as contrasted against unloaded samples. Finally, it can be reported that the gradual growth of *PP* in RD and TD for MC1100 depicted in Figure 8 can be found approximately at the same tensile stress (about 400 MPa) as compared with the knee in the evolution of MBN shown in Figure 7. In addition, the saturation of MBN in TD seems to coincide with the saturation of *PP* in TD at higher stresses.

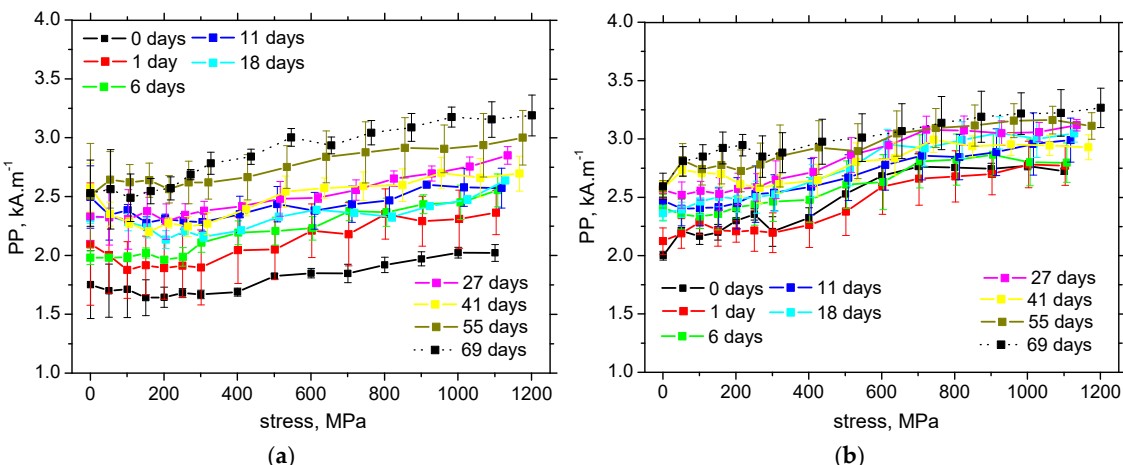

**Figure 8.** Evolution of *PP* versus tensile stress for MC1100. (**a**) RD, (**b**) TD.

A quite interesting evolution of the *FWHM* of the MBN envelope can be reported with respect to the different microstructures of the investigated steels. The *FHWM* for all steels grows along with a more developed corrosion attack (see Figure 9). However, the *FWHM* for S235 drops down (as reported earlier [29]). The *FWHM* for MC700 drops only gently, or a flat evolution occurs, and finally, this evolution for MC1100 becomes fully reversed.

After an initial, very gentle decrease, the *FWHM* for MC1100 exhibits moderate growth (see Figure 9). It should be reported that the sensitivity of *FWHM* against corrosion attack is quite good. However, a lack of sensitivity against the tensile stress can be obtained for MC700 and MC 1100 as contrasted against a valuable decrease in *FWHM* with tensile stress for S235.

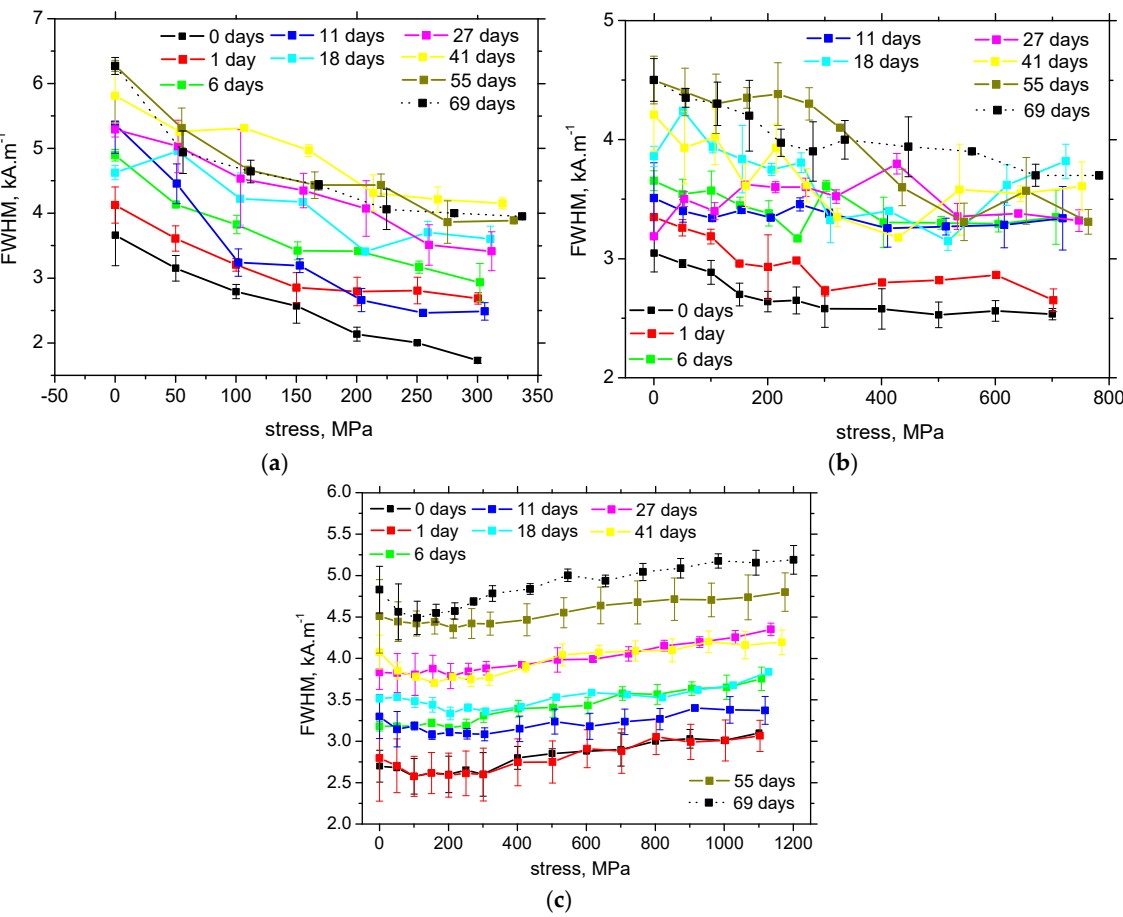

**Figure 9.** Evolution of *FWHM* versus tensile stress in RD. (**a**) S235, (**b**) MC700, (**c**) MC1100.

The increase in *FWHM* along with corrosion attack is attributed to the increasing scale (size) of ferromagnetic domains and the corresponding DWs contributing to MBN. The low-amplitude MBN pulses at higher magnetic fields originating from the small un-corroded fragments in the corroded layer are mixed with the larger MBN pulses occurring at the lower magnetic fields originating from the larger corroded particles and/or the underlying uncorroded matrix [14,15]. More developed corrosion increases the diversity of MBN sources, which in turn increases *FWHM*.

The different evolution of *FWHM* for S235 compared with MC700 and especially MC1100 is driven by the different microstructures of the investigated steels. The SEM as well as metallographic images of the different microstructures of the investigated steels can be found in the previous study [35]. S235 is composed of equiaxed grains in which demagnetizing fields in different directions are nearly the same. Tensile stress aligns DWs in the direction of this stress, and this alignment becomes more developed at higher tensile stresses, which in turn decreases *FWHM* (conditions for DW's irreversible motion become more similar along their similar directions). However, MC700 is a mixture of equiaxed ferritic grains and plate bainite, and MC1100 is fully composed of plate martensite [16]. The plate characteristics of bainite and martensite preferentially align DWs along the length of these plates as a result of demagnetizing fields [18,19]. Preferential alignment of the DWs in the unloaded state consumes the aforementioned realignment of the DWs during tensile stressing. For this reason, the *FWHM* for MC1100 exhibits a reverse evolution as

compared with S235. The evolution of *FWHM* for MC700 is, therefore, a mixture of those for S235 and MC1100 since the microstructure is also a mixture of equiaxed ferritic grains and plate bainite.

The collection of information with respect to the influence of corrosion as well as the superimposing contribution of the stress state is quite complex. For this reason, the summing contribution of corrosion and the stress state on the extracted MBN features is listed in Table 2.

**Table 2.** Evolution of MBN parameters as a function of corrosion damage and the superimposing tensile elastic stresses.

| | MBN—Effective Value | | *PP* | | *FWHM* | |
|---|---|---|---|---|---|---|
| | **RD** | **TD** | **RD** | **TD** | **RD** | **TD** |
| corrosion | ↓ | ↓ | ↑ | ↑ | ↑ | ↑ |
| Tensile stress | ↑ | ↓ | ↑ | ↑ | ↓ * | ↓ * |

↑—ascending evolution; ↓—descending evolution; *—flat evolution of MC1100.

Table 2 clearly demonstrates the different evolution of the different MBN parameters extracted from the filtered MBN signals. Corrosion attack results in the growth of *FWHM*, which is compensated by the decreasing evolution along with the tensile stresses. The effective values drop down with the higher extent of corrosion damage. However, the response with respect to the tensile stress is asymmetric in RD and TD due to realignment of the DWs into RD. Finally, *PP* tends to increase with the corrosion attack as well as the tensile stress and this parameter only exhibits the systematic behavior in RD as well as TD. On the other hand, MBN-extracted parameters as well as their combination provide no exclusive values on which the pure contribution of corrosion and tensile stress can be distinguished.

## 4. Conclusions

A corroded layer covering an un-corroded body makes the assessment of the stress state more difficult (when compared with the components being un-corroded). MBN is sensitive against the rolling direction of low-alloyed steel, the degree of corrosion attack, and the direction of measurements with respect to the direction of the exerted stress as well as the amplitude of the stress. Synergistic contributions of these aspects do not enable distinguishing among them when only the MBN value is employed. MBN values for all investigated low-alloyed steels are not exclusive for any mixture of the aforementioned aspects, and they overlap with respect to the variable stress state, corrosion attack, and/or the direction in which the MBN measurement occurs. For this reason, monitoring components with respect to their corrosion damage and superimposing stress should be based on a mixture of MBN parameters extracted from the raw MBN signals. This can be summarized as follows:

- Higher MBN values are linked with lower thicknesses of a corroded layer;
- Higher MBN values are linked with higher tensile stresses in RD and a lower MBN in TD;
- *PP* values in RD are higher for heavily corroded samples, but the sensitivity against stress is limited;
- *FWHM* values in RD are higher for heavily corroded samples, but the sensitivity against stress for MC700 and MC1100 is also limited;
- *FWHM* for S235 drops down with tensile stress in RD;
- *PP* and *FWHM* in TD exhibit a lack of sensitivity against corrosion attack as well as tensile stress.

Finally, it should be considered that the multi-parametric approach would be beneficial in the proposal of a suitable concept when all sensitive MBN parameters, such as MBN, *PP*, and *FWHM*, are employed.

**Author Contributions:** Conceptualization, M.N. and F.B.; methodology, F.P. and T.K.; software, R.K. and T.K.; validation, F.B.; formal analysis, R.K.; investigation, M.N., F.P., R.K. and F.B.; resources, F.B. and T.K.; data curation, R.K. and F.P., writing—original draft preparation, M.N. and F.P.; writing—review and editing, M.N.; visualization, R.K. and T.K.; supervision, F.P. and F.B.; project administration, F.B.; funding acquisition, F.B. All authors have read and agreed to the published version of the manuscript.

**Funding:** This publication was realized with support from the Operational Program Integrated Infrastructure 2014–2020 of the project: Innovative Solutions for Propulsion, Power and Safety Components of Transport Vehicles, code ITMS 313011V334, co-financed by the European Regional Development Fund. Additionally, support of the VEGA project 1/0052/22 is also gratefully acknowledged. This research was also funded by Operational Program Integrated Infrastructure: Application of innovative technologies focused on the interaction of engineering constructions of transport infrastructure and the geological environment, ITMS2014+ code 313011BWS1. The project was co-funded by the European Regional Development Fund.

**Institutional Review Board Statement:** Not applicable.

**Informed Consent Statement:** Not applicable.

**Data Availability Statement:** The raw data required to reproduce these findings cannot be shared easily, due to technical limitations (some files are too large). However, authors can share the data on any individual request (please contact the corresponding author by the use of their mailing address).

**Conflicts of Interest:** The authors declare no conflict of interest.

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
