# Peer review of "Barkhausen Noise Emission as a Function of Tensile Stress in Low-Alloyed Steels: Influence of Corrosion and Steel Strength"

_applsci, doi:10.3390/app13116574_

Round 1
Reviewer 1 Report
The article analyzes the correlation between Barkhausen noise emission and tensile stress, attempting to provide a non-destructive method for detecting tensile stress. The theme of this article is good and has certain engineering significance. However, the article only describes the phenomenon without theoretical analysis. In addition, the microstructure analysis mentioned in the article has almost no relevant content, and the conclusions lack a basis. Some comments are as follows:
1. There is no Part 4, proceed directly to Part 5.
2. The article specifies that the sample MC1100 is 6 mm thick, while S235 and MC700 are 5 mm thick. The thickness of the three samples should be consistent.
3. Figure 3 gives the percentage reduction in effective wall thickness. Since the thickness is not consistent, the conclusions drawn from this are unreasonable.
4. According to Figure 5, the conclusion "MBN in RD is greater as compared with TD" is inaccurate when the tensile stress is relatively small.
5. Comparing Figure 5(b) and Figure 6(b), it can be seen that when the tensile stress is relatively high, the MBN of MC700 in the TD direction is greater than that of MC1100, thus the conclusion "MBN for MC700 is less than MC1100" should be reconsidered.
6. The aim of the article is to investigate the use of Barkhausen noise for non-destructive testing and to mention that it can detect the degree of corrosion. The article analyzes the changes in three parameters of Barkhausen noise in different tensile stress, corrosion degree, and material yield strength but does not explain how to detect the degree of corrosion.
Minor editing of English language required
Author Response
All changes made in the manuscript (additional texts and corrections) are highlighted yellow colour (valid for the manuscript as well as this document).
Reviewer: There is no Part 4, proceed directly to Part 5.
Response: we agree
Manuscript: corrected
Reviewer: The article specifies that the sample MC1100 is 6 mm thick, while S235 and MC700 are 5 mm thick. The thickness of the three samples should be consistent.
Response: We agree. However, it was impossible to purchase to sheet of the MC1100 from supplier of the thickness 5 mm. For this reason, we carried out the measurement with this thickness. The different samples thickness no influence of MBN due to limited sensing depth of MBN.
Manuscript: We prefer no change.
Reviewer: Figure 3 gives the percentage reduction in effective wall thickness. Since the thickness is not consistent, the conclusions drawn from this are unreasonable.
Response: we disagree. The percentage reduction is reasonable due to the different samples thickness. On the other hand, the different samples thickness does not have influence on the thickness of corroded layer. Only the true stress state has to be recalculated with respect of the different thickness of the samples.
Manuscript: We prefer no change.
Reviewer: According to Figure 5, the conclusion "MBN in RD is greater as compared with TD" is inaccurate when the tensile stress is relatively small.
Response: We agree.
Manuscript: We added the comment.
MBN in RD is mostly greater as compared with TD. Only in the region of the lower magnitudes of the tensile stresses MBN values are more balanced and the magnetic anisotropy expressed in term of MBN is lost.
Reviewer: Comparing Figure 5(b) and Figure 6(b), it can be seen that when the tensile stress is relatively high, the MBN of MC700 in the TD direction is greater than that of MC1100, thus the conclusion "MBN for MC700 is less than MC1100" should be reconsidered.
Response: We agree.
Manuscript: We added the comment.
On the other hand, it can be seen that MBN in TD for MC700 are more as those for MC1100 at the higher tensile stresses especially for the heavily corroded samples, see Figure 6b and Figure 7b.
Reviewer: The aim of the article is to investigate the use of Barkhausen noise for non-destructive testing and to mention that it can detect the degree of corrosion. The article analyzes the changes in three parameters of Barkhausen noise in different tensile stress, corrosion degree, and material yield strength but does not explain how to detect the degree of corrosion.
Response: This topic was reported in our earlier study [16]. The pure contribution of only corrosion on MBN is also integrated in this study. However, the main focus can be found the superimposing contribution of corrosion and stress state.
Manuscript: We prefer no change. Please check our previous study [16].
Reviewer 2 Report
In this study, tensile stress in low alloyed steels is non-destructive monitored using the Barkhausen noise emission method. Authors have done many works and obtained good results. But there are some parts should be carefully modified or added before this work is further consideration.
(1) The experiments are unclear. Please provide the diagrammatic sketch or equipment setups for the work. Which signals are used and how to extract the useful signals. These important information is lost.
(2) Authors provide many results to discuss. What is the final results or conclusion? As the title mentions, what is the function of tensile stress affected by the other information? There lacks some clear and concise conclusions, expressed using figures or equations.
Author Response
All changes made in the manuscript (additional texts and corrections) are highlighted yellow colour (valid for the manuscript as well as this document).
Reviewer: The experiments are unclear. Please provide the diagrammatic sketch or equipment setups for the work. Which signals are used and how to extract the useful signals. These important information is lost.
Response: we added the required diagrammatic sketch.
Manuscript: we added the text. We also added new Figure 2.
The brief diagrammatic sketch of equipment and extracted information for easier navigation through the paper is presented in Figure 2.
Reviewer: Authors provide many results to discuss. What is the final results or conclusion? As the title mentions, what is the function of tensile stress affected by the other information? There lacks some clear and concise conclusions, expressed using figures or equations.
Response: We agree. Therefore, we added new Table 2 as well as additional comments.
Manuscript: Please check new Table 2. We also added new text.
The collection of information with respect of the influence of corrosion as well as the superimposing contribution of stress state is quite complex. For this reason, the summing contribution of corrosion and stress state on the extracted MBN features is listed in Table 2.
Table 2 clearly demonstrates the different evolution of the different MBN parameters extracted from the filtered MBN signals. Corrosion attack results into the growing FWHM which is compensated by the decreasing evolution along with the tensile stresses. The effective values drop down with the higher extent of corrosion damage. However, the response with respect of the tensile stress is asymmetric in RD and TD due to realignment of DWs into RD. And finally, PP tends to increases with the corrosion attack as well as the tensile stress and this parameter only exhibits the systematic behaviour in RD as well as TD. On the other hand, MBN extracted parameters as well as their combination provide no exclusive values on the base which the pure contribution of corrosion and tensile stress can be distinguished.
Reviewer 3 Report
This paper studied influence of corrosion and steel strength in low 2 alloyed steels by Barkhausen noise emission as a function of tensile stress. However, some important experiments should be added for evidence. It is recommended that this manuscript is possibly considered after making major revision taking into account the following remarks. Some other errors are also necessary to be corrected.
1. In the abstract, it mainly focuses on macroscopic description, but lacks quantitative conclusion. Hence it is suggested to improve the abstract to include some important conclusions.
2. In Figure 2, it is difficult to distinguish change of morphology before and after corrosion attack. It is better to supply microscopic images (eg. SEM), and mark the surface changes in the images.
3. The authors mentioned ‘The different evolution of FWHM for S235 compared with MC700 and especially 320 MC1100 is driven by the different microstructures of the investigated steels.’ However, there is no evidence in this paper. The authors should supply SEM images of different steels, and characterize the microstructure change before and after tensile stress.
4. The paper lacks of discussion of mechanism, and it is suggested to combine all the data for further analysis.
5. There are too much description in the conclusions, but still unclear. The conclusive description should be added. Please modify it.
Minor editing of English language required.
Author Response
All changes made in the manuscript (additional texts and corrections) are highlighted yellow colour (valid for the manuscript as well as this document).
Reviewer: In the abstract, it mainly focuses on macroscopic description, but lacks quantitative conclusion. Hence it is suggested to improve the abstract to include some important conclusions.
Response: we added the required information into the abstract.
Manuscript:
Corrosion attack results into the growing FWHM which is compensated by the decreasing evolution along with the tensile stresses. The effective values drop down with the higher extent of corrosion damage. However, the response with respect of the tensile stress is asymmetric in RD and TD due to realignment of DWs into RD. And finally, PP tends to increases with the corrosion attack as well as the tensile stress and this parameter only exhibits the systematic behaviour in RD as well as TD. On the other hand, MBN extracted parameters as well as their combination provide no exclusive values on the base which the pure contribution of corrosion and tensile stress can be distinguished.
Reviewer: In Figure 2, it is difficult to distinguish change of morphology before and after corrosion attack. It is better to supply microscopic images (eg. SEM), and mark the surface changes in the images.
Response: These images can be found in our previous study [16] as well as the further information about the thickness of corroded layer. The images presented in [16] clearly demonstrate the difference between the corroded layer as well as uncorroded matrix. Moreover, SEM images of corroded layer together with chemical analysis can be found in [14].
Manuscript: We added links to the studies in which the required information can be found.
Metallographic details in which the corroded layer can be distinguished from unaffected matrix can be found in the previous study [16]. SEM observations as well as the chemical analyses of the corroded layer was also reported earlier [14].
Reviewer: The authors mentioned ‘The different evolution of FWHM for S235 compared with MC700 and especially 320 MC1100 is driven by the different microstructures of the investigated steels.’ However, there is no evidence in this paper. The authors should supply SEM images of different steels, and characterize the microstructure change before and after tensile stress.
Response: We agree. The microstructures, metallographic as well as SEM observations can be found in our previous study. For this reason, we added the link to this study in order to see the difference in microstructures.
Information about the changes before and after the tensile test is not relevant since the stress state is in the elastic region only. Therefore, no permanent changes of microstructure are expected.
Manuscript:
The SEM as well as metallographic images of the different microstructures of the investigated steels can be found in the previous study [35].
- Neslušan, M.; Pitoňák, M.; Minárik, P.; Tkáč, M.; Kollár, P.; Životský, O. Influence of domain walls thickness, density and alignment on Barkhausen noise emission in low alloyed steels, Rep. 2023, 13 5687; doi: 10.1038/s41598-023-32792-1.
Reviewer: The paper lacks of discussion of mechanism, and it is suggested to combine all the data for further analysis.
Reviewer: There are too much description in the conclusions, but still unclear. The conclusive description should be added. Please modify it.
Response: We agree with the second comment. Therefore, we added new Table 2 as well as additional comments. On the other hand, we provided quite rich explanation of mechanism in which corrosion s well as stress state affects MBN.
Manuscript: Please check new Table 2. We also added new text.
The collection of information with respect of the influence of corrosion as well as the superimposing contribution of stress state is quite complex. For this reason, the summing contribution of corrosion and stress state on the extracted MBN features is listed in Table 2.
Table 2 clearly demonstrates the different evolution of the different MBN parameters extracted from the filtered MBN signals. Corrosion attack results into the growing FWHM which is compensated by the decreasing evolution along with the tensile stresses. The effective values drop down with the higher extent of corrosion damage. However, the response with respect of the tensile stress is asymmetric in RD and TD due to realignment of DWs into RD. And finally, PP tends to increases with the corrosion attack as well as the tensile stress and this parameter only exhibits the systematic behaviour in RD as well as TD. On the other hand, MBN extracted parameters as well as their combination provide no exclusive values on the base which the pure contribution of corrosion and tensile stress can be distinguished.
Reviewer 4 Report
A brief summary:
The aim of the paper was to investigate the influence of corrosion and steel strength on Barkhausen noise in low alloyed steels. The paper constitutes a continuation of the research on the Barkhausen noise as a non-destructive technique to monitor the extent of corrosion and its influence on construction stability. The work is applicable in civil engineering as it expands the knowledge on alternative corrosion monitoring techniques. Authors conducted experimental tests on the relation of tensile stress and Barkhausen noise emission for different samples of 3 types of steels. The results prove the relation between the Barkhausen noise emission levels and the corrosion state of steel samples and thus indicate a possible industrial application.
Broad comments:
A significant achievement and main novelty of the presented work is the investigation of the relation between the Barkhausen noise emission and the tensile stress levels in low alloyed steels. The subject of the paper is practical and important as it gives more data on possible application of Barkhausen noise as a monitoring tool of corrosion state of civil engineering steels. The authors conducted experimental tests that showed a good correlation between the MBM and strength variations in low alloyed steels due to corrosion. The introduction of the paper provides sufficient background on the Barkhausen noise as a monitoring signal. However, other monitoring techniques, such as acoustic emission, ultrasonic tomography, gamma rays or electromagnetic signals have only been mentioned. Nothing in the introduction indicate what is the advantage of using Barkhausen noise in relation to other methods. The article would greatly benefit if some additional info in the Introduction was supplemented. The literature references are numerous, relevant and up to date. The methods and results are presented in a clear way. The conclusions that the authors draw from the experiment tests are sufficiently detailed and supported by the results. The language and editing of the paper are fine, although there are some minor errors that should be corrected.
Specific comments:
· No info concerning why the Barkhausen noise should be consider as a monitoring method in relation to other, established methods.
· Why authors haven’t applied in the experimental research any Design of Experiment tool?
· Figures 4, 5 and 6 are in the paper before any reference to them in the text is made.
The language of the paper is fine, although there are some minor errors that should be corrected, i.e. line 66 “Increasing surface irregularities make the propagation…” etc.
Author Response
All changes made in the manuscript (additional texts and corrections) are highlighted yellow colour (valid for the manuscript as well as this document).
Reviewer: No info concerning why the Barkhausen noise should be consider as a monitoring method in relation to other, established methods.
Response: we added further explanation.
Manuscript: We added this text at the end of Introduction.
MBN as a monitoring technique is very fast and reliable. Previous experience demonstrated its good sensitivity against the developed corrosion extent [16]. Furthermore, MBN is sensitive to the stress state [24, 28]. For these reasons, this technique was employed in this study for the aforementioned purpose.
Reviewer: Why authors haven’t applied in the experimental research any Design of Experiment tool?
Response: We carried out our investigation in the simplified mode due to complicated nature of MBN. In this study the contribution of the different materials is mixed with corrosion as well as its influence on stress state. Moreover, the stress state is investigated in 2 different directions. The problem is therefore very complex. Therefore, we prefer to propose and keep the experiments in the form presented in the initial version of the study.
Manuscript: We prefer no change.
Reviewer: Figures 4, 5 and 6 are in the paper before any reference to them in the text is made.
Response: We agree. Therefore, we altered the sequence of the text. The text in which these figures are mentioned are before the appearance of the linked figures.
Manuscript: please check it in the text.
Round 2
Reviewer 2 Report
This work can be accepted as it is.
Reviewer 3 Report
This manuscript can be accepcted in present form.